# Modeling chronic wasting disease transmission risk in mule deer related to habitat characteristics

Erica M. Christensen[1*¤a], Nathan J. Kleist[1], David R. Edmunds[1], Julie A. Heinrichs[2], D. Joanne Saher[2], Ashley L. Whipple[1], Melia DeVivo[3¤b], Cameron L. Aldridge[1]

**1** United States of America Geological Survey, Fort Collins Science Center, Fort Collins, Colorado, United States of America, **2** Natural Resource Ecology Laboratory, Colorado State University, Colorado, United States of America, **3** Department of Veterinary Sciences, University of Wyoming, Laramie, Wyoming, United States of America

¤a Current Address: Department of Fish, Wildlife, and Conservation Ecology, New Mexico State University, Las Cruces, New Mexico, United States of America
¤b Current Address: Washington Department of Fish and Wildlife, Spokane Valley, Washington, United States of America
* echriste@nmsu.edu

## Abstract

Chronic wasting disease (CWD) is a prion disease of cervids that spreads to uninfected individuals through direct transmission (contact with infected individuals), vertical transmission (from mother to offspring), or indirect transmission (exposure to contaminated environments). The risk of indirect transmission is unevenly distributed on the landscape, and risk levels are expected to be controlled by patterns of habitat use by infected and uninfected individuals as well as environmental properties that alter the length of time prions remain infectious and available for uptake. Despite evidence from controlled or laboratory studies identifying environmental properties likely to affect patterns of CWD prion locations on the landscape, it remains difficult to connect mechanisms to realized increased or decreased risk of disease transmission, and few studies have attempted to detect patterns of different CWD risk in different environments. Using data from GPS-collared mule deer in Wyoming that were CWD-tested annually, we constructed models predicting annual probability of disease transmission contingent on environmental properties extracted from GPS use points. We compared models that emphasized different pathways of disease transmission by including or excluding sets of covariates that described deer density, habitat selection, and covariates expected to affect prion persistence in the environment. Results indicated that key habitat characteristics often selected by mule deer, such as proximity to secondary roads, were also associated with higher risk of testing positive for CWD, which supports the hypothesis that disease risk was correlated to patterns of habitat use by deer. We also found increased risk associated with spatial properties that were not selected-for by deer, such as areas where topography collects moisture, suggesting that prion retention mechanisms also play a role in risk. Incorporating these spatially-varying risk factors into our understanding of CWD transmission

**Data availability statement:** Mule deer capture data used as input to CWD risk models are available from Dryad (DOI https://doi.org/10.5061/dryad.h66cn). GPS locations cannot be shared publicly because they are considered sensitive information, but can be obtained by request from Movebank (movebank.org, study name "Chronic Wasting Disease Ecology and Epidemiology of Mule Deer in Wyoming," study ID 7727287671). All environmental covariate data are publicly available as cited in the manuscript methods. Risk maps produced as output from these analyses are available as raster files within a publicly available data release (doi.org/10.5066/P1K3QFC8).

**Funding:** This work was supported by internal funding awarded to JH, DE, and CA from the U.S. Geological Survey, Biological Threats and Invasive Species Research Program. Sponsors did not play a role in study design, data collection and analysis, decision to publish, or preparation of the manuscript.

**Competing interests:** The authors have declared that no competing interests exist.

and outbreak progression can support managers in designing data collection and disease management strategies.

## Introduction

Chronic wasting disease (CWD) is a transmissible spongiform encephalopathy, or prion disease, that can have devastating effects on cervid populations [1,2]. In North America, CWD is found in wild populations of mule deer (*Odocoileus hemionus*), white-tailed deer (*Odocoileus virginianus*), elk (*Cervus elaphus*), and moose *(Alces alces)*. Given the high costs of monitoring and managing populations affected by CWD [3], understanding the factors that contribute to disease transmission risk can be a high priority for wildlife managers. Chronic wasting disease is spread directly, through animal-to-animal contact, vertically, from mother to offspring, and indirectly, when uninfected individuals are exposed to infectious prions that were deposited in the environment by infected individuals. Disease transmission through direct contact is influenced by social dynamics [4], and transmission is likely to be higher in locations where cervids tend to congregate [5]. Evidence of prions in fetal and reproductive tissues across several cervid species suggests that transmission from mother to offspring can occur during pregnancy, even before birth or direct contact [6,7]. Rates and probability of disease transmission through indirect mechanisms are less well understood. Prion exposure from environmental sources is difficult to measure, since risk from the environment is intermingled with herd social behavior and habitat selection and use patterns. Nevertheless, indirect disease transmission is an important source of disease spread [8].

Indirect transmission occurs when a susceptible host ingests infectious prions from the environment and subsequently develops CWD. The potential for environment-based infection is not evenly distributed on the landscape; prion load, i.e., the presence and number of infective prions within a habitat patch, is affected by the rate of prion deposition by infected individuals and abiotic environmental properties that affect the ability of prions to remain intact and available for uptake. Habitats that are selectively used by host species will likely have higher prion loads, as infective prions are deposited by infected individuals in urine, feces, saliva, antler velvet, blood, and in the carcasses of dead individuals [9]. Once deposited, prions can remain in an infective state for extended periods of time [9–11], and the length of time they remain on environmental substrates and retain infectivity varies based on different substrate properties. Prions can bind to plant material, and may even be taken up by plants from the soil and incorporated into tissues [12,13]. Prions can also be transported by water and have been detected in natural water sources in a CWD-endemic area [14]. Many studies have explored interactions between prions and soil properties. Prions bind to clay particles in soil, therefore clay-rich surface soils are expected to keep infective particles close to the surface and available for uptake [15], and binding with soil particles may enhance infectivity of prions [16]. Prion infectivity and availability are also expected to be affected by soil organic content [17], pH [18], and percent sand content [15]. Repeated freeze-thaw cycles damage prion

structures and reduce their ability to cause infection [19]; however, this effect was reduced when prions were dehydrated prior to freezing [20], suggesting a complex relationship between duration of prion infectivity and combined moisture and temperature conditions.

Despite knowledge of how specific soil and environmental properties affect CWD prions, it remains difficult to predict or quantify how these properties contribute to increased or decreased disease transmission risk on the landscape. Studies aimed at evaluating risk based on environmental properties have suggested patterns such as higher risk in areas with high stream density and agricultural fields [21,22], lower risk in urban areas [23], and higher risk associated with higher soil pH [24]. However, risk factors may be highly specific to geographic location, as indicated by occasionally contradictory evidence from different studies: higher soil clay content was associated with higher CWD risk in Colorado [25], but higher clay content was associated with lower CWD risk in Illinois and Wisconsin [23,24]; higher CWD risk was associated with areas with lower forest cover in West Virginia [26], but higher CWD risk was found in areas with large, continuous forest patches in Illinois and Wisconsin [23]. The question of how habitat properties contribute to CWD risk is often confounded by the fact that higher CWD prevalences are found in areas with higher host species density [21,23].

We approached the problem of detecting correlations between environmental properties and CWD transmission risk using a dataset from global positioning system (GPS) collared mule deer (*Odocoileus hemionus*) in a CWD endemic area of Wyoming [2]. Because CWD had been present in this study area for at least four decades at the time of the field study [27–29], we expected environmental prion loads to be substantial and therefore indirect transmission was more likely to be an important mechanism of CWD transmission than would be observed in an area without a long history of CWD presence [4,30]. Collared deer were captured and antemortem tested annually for CWD via tonsil biopsy and immunohistochemistry (IHC) testing. These data allowed us to identify individual deer that likely contracted CWD within a one-year time span: i.e., deer that tested positive for CWD after a non-detection test the previous year. We compared patterns of habitat use by these likely newly infected deer to patterns of use by deer that did not test positive over the same time span. This unique dataset provides a more detailed estimation of individual habitat use than some previous studies linking habitat to CWD risk, which have extracted environmental characteristics from an assumed home range polygon around hunter kill locations of infected and uninfected deer [23,25], or others that have examined effects of environmental disease transmission based on captive animals in controlled settings [31]. Furthermore, we were able to use the deer location dataset to model resource selection by mule deer in this herd. This allowed us to construct models correlating CWD transmission risk to a measure of relative habitat usability, for comparison with models correlating risk to environmental characteristics independent of deer selection patterns.

We modeled the probability of an individual transitioning from an uninfected to infected state in a one-year period, contingent on the properties of the habitat used by the individual during that period, using generalized linear models (GLMs) with a binomial response. We constructed models based on five hypotheses about which disease transmission mechanisms may be dominant in this system: 1) null hypothesis, in which disease risk is driven by non-environmental factors only (e.g., age, genotype of the prion protein gene [*PRNP*]); 2) deer density hypothesis, in which risk is assumed to be concentrated in geographic areas most frequently used by deer, as measured by density of GPS use points on the landscape; 3) habitat suitability hypothesis, in which risk is also assumed to be related to deer habitat use, but risk is more strongly correlated to covariates that affect habitat selection (e.g., presence of roads, agriculture) than to direct measures of deer density; 4) prion persistence hypothesis, in which risk is assumed to be driven by environmental covariates that affect the persistence of prions on the landscape (e.g., soil clay content); and 5) combination hypothesis, in which covariates representing deer density, habitat suitability, and prion persistence are all important. Our goal with these analyses was to identify specific environmental properties correlated with increased or decreased probability of deer testing positive for CWD. Though our main motivation was to leverage these correlations to predict patterns of CWD transmission risk on the landscape rather than explain causal relationships driving these patterns [32], our results can also inform predictions about the mechanisms driving CWD risk in this area that can guide future research.

## Materials and methods

### Study area and data collection

Data used in these analyses were collected as part of a study to investigate population dynamics and CWD prevalence in the South Converse Mule Deer Herd (SCMDH) located in Converse County, Wyoming (Fig 1) [2]. The study area consisted of mainly private native rangelands with some cultivated meadows and tracts of public land. Deer wintered at elevations ~1,500 m and a portion of the population migrated to summer ranges at ~2,700 m, where larger tracts of national forest occur. Mule deer were captured annually in February or March from 2010 to 2014. Data collection methods and procedures were approved by University of Wyoming Institutional Animal Care and Use Committee (No. A-3216-01) and the Wyoming Game and Fish Department (Permit No. 33–751). Each deer was fitted with a store-onboard GPS radio collar (Lotek Wireless Inc., Newmarket, Ontario, Canada) at initial capture, which recorded locations every 6 or 8 hours. Blood was collected from each deer to determine codon 225 *PRNP* genotype, and each deer was characterized as either "SS" (homozygous for serine at codon 225) or "*F" (either heterozygous or homozygous for phenylalanine at codon 225, "SF" or "FF"). We grouped deer with SF and FF genotypes together in this analysis because only one individual in our dataset was FF, and it is well-established in the literature that deer possessing at least one F allele at codon 225 are less likely to test positive for CWD and those individuals that do acquire CWD experience slower disease progression than the "SS" type [33,34]. Deer age was estimated from tooth eruption and wear by a team of experienced observers [35], and confirmed via examination of cementum annuli of an extracted tooth in the event of deer mortality. Only yearling (1.5-year old) and older deer were captured. At each annual capture, tonsil biopsies were performed to test for CWD using IHC, providing an annual record of CWD status for each individual. Surgical equipment were cleaned and disinfected using methods previously published to prevent iatrogenic transmission of CWD [36]. In the event of a deer mortality, carcasses

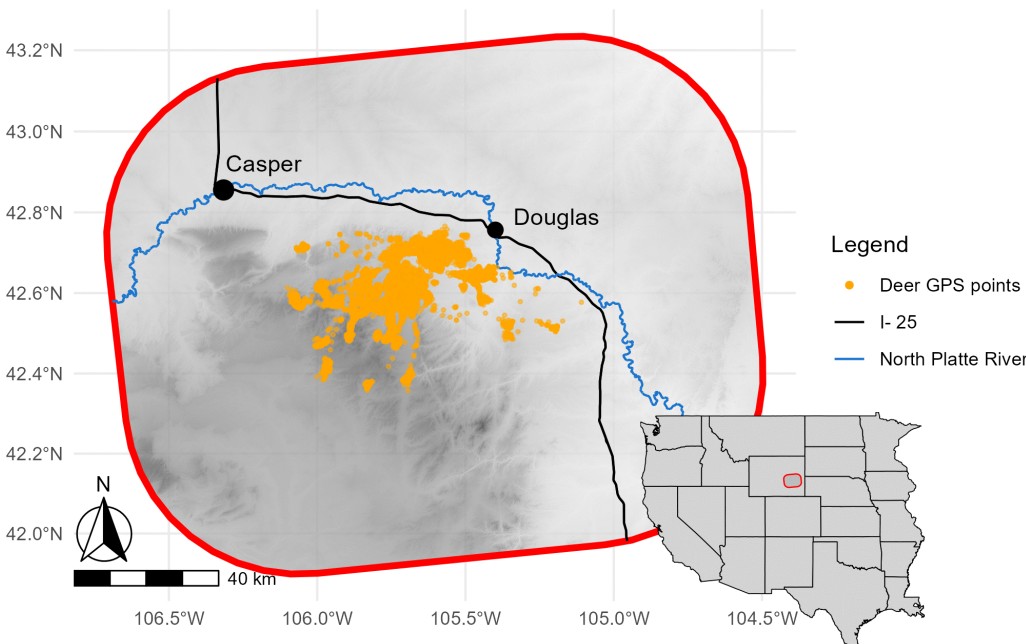

**Fig 1. South Converse (Wyoming, USA) study area displaying GPS points from 122 collared mule deer.** Gray shading indicates elevation, with darker gray indicating higher elevation. The towns of Casper and Douglas are indicated with black circles. Data for elevation, road, and river were obtained from publicly available data sources [38–40]. Map was created using R software [41] including packages 'ggplot2,' 'ggspatial,' 'maps,' 'cowplot,' and 'terra' [42–46].

were recovered and a postmortem CWD test was performed on retropharyngeal lymph nodes, tonsil, and/or the obex region of the medulla oblongata. Refer to [2] for detailed data collection methods. Data on each deer's sex, genotype, age at first capture, CWD test result at first capture, date of last test with CWD not detected, and date of first positive CWD test result (if applicable) are publicly available on Dryad [37]. The GPS locations are available by request from Movebank (movebank.org, study name "Chronic Wasting Disease Ecology and Epidemiology of Mule Deer in Wyoming," study ID 7727287671).

## Disease transmission risk models

We modeled the probability a deer transitioned from a CWD non-detection to a positive CWD test in a 1-year period based on habitat use by the deer during that period using GLMs. The response variable was disease outcome, value 1 or 0 indicating whether or not the deer transitioned from a non-detection CWD test to a positive CWD test, suggesting it contracted CWD during that time. We included only female deer in these models, as females were the main target of the original study [2] and sample size of males was too small for analyses. We excluded deer that tested positive for CWD at first capture, deer that did not have at least one year of GPS data (which we defined as at least 300 days with recorded GPS points, approximately 900 GPS points) with a before and after CWD test, and deer with collar malfunctions that resulted in large temporal gaps in GPS data. Non-detection CWD follicle tests where <6 follicles were collected were considered unreliable due to an increased chance of a false negative [47]; we therefore excluded these samples from our CWD test dataset unless there was additional evidence (i.e., a non-detection test in a later year with > 6 follicles or a non-detection postmortem test implied that all previous tests were likely negative). This resulted in a dataset containing 59 female mule deer, each with 1–4 years of GPS data and CWD testing records. Though our dataset included multiple years of records from some individual deer, we did not include individual as a random effect in models because the majority of individuals (38 of 59) only had one year of data (S1 Table).

We constructed models based on our five hypotheses about which disease transmission mechanisms may be dominant in this system: 1) null hypothesis, 2) deer density hypothesis, 3) habitat suitability hypothesis, 4) prion persistence hypothesis, and 5) combination hypothesis. For each hypothesis, we constructed one or more GLMs containing covariates related to the focal mechanism and used Akaike's Information Criterion for small sample sizes ($AIC_c$) to select the most parsimonious models [48].

Many of the covariates explored were spatial properties representing environmental conditions at the GPS points visited by deer. We extracted spatial covariate values at each GPS point for the 59 deer included in the transmission risk models and aggregated to a yearly value. For continuous covariates (e.g., percent tree cover), values were aggregated by calculating the average for each deer and year. Year was defined as the dates in between annual CWD testing events. For categorical values, (e.g., National Land Cover Database [NLCD] land cover types [49]), we extracted values at each GPS point and calculated the percent of each deer's yearly use points spent in each category to obtain a measure of percent time spent in the land type category (e.g., "percent time spent in wetland"). We combined several NLCD categories that were closely related; "developed, high intensity," "developed, medium intensity," and "developed, low intensity" were combined into a single "developed" category, and "woody wetlands" and "emergent herbaceous wetlands" were combined into a single "wetland" category. Additional details on processing of spatial environmental data can be found in S1 File.

We explored the effect of spatial covariates at different scales by extracting covariate data using three buffer sizes around GPS points: buffer size of 0m (single pixel extraction from rasters), 50m (which corresponds to a 3x3 pixel neighborhood), and 500m. The average distance traveled in an 8-hour period by deer in this data set was approximately 500m, so a radius of 500m around each GPS point represents maximum possible habitat use. At each scale, we summarized spatial covariates by deer and year. We found that the covariates at the three scales were highly correlated to each other, and single-variable risk models containing covariates at the three scales had very similar support when compared using $AIC_c$. We therefore restricted subsequent analyses to covariates extracted at the pixel level.

Additionally, certain habitat characteristics were expected to have a different relationship with infection risk at different times of year. For example, north-facing aspect values in winter likely represent seasonally frozen soils, and freezing is known to affect prion persistence and infectivity [19]. We therefore calculated average covariate values by each season (summer/winter) as well as annually. Winter was defined as October 1 to April 30, which roughly corresponds to first and last frost dates in the region.

### Hypothesis 1: null model

Under the null model hypothesis (hypothesis 1), we tested for correlations between risk of CWD transmission and covariates not related to habitat used by deer. We constructed GLMs including non-environmental variables expected to affect probability of infection: *PRNP* genotype, deer age, observation year, and whether the deer was migratory or nonmigratory. Any of these variables found to be significant predictors of CWD risk were subsequently included in models testing hypotheses based on environmental predictors, in order to control for important non-environmental factors.

### Hypothesis 2: deer density hypothesis

Under our deer density hypothesis, risk of CWD transmission was expected to be greater in locations more heavily used by deer, both as a result of heavier deposition of infectious prions in feces and other fluids and tissues, as well as higher chance of direct deer-to-deer contact. To test this hypothesis, we used the results of a kernel density estimate (KDE), an analysis to quantify spatial "hotspots" of high deer use in our study area, as a predictor of risk. The KDE analysis was performed using ArcGIS Spatial Analyst toolbox (Copyright 2024 Esri Inc., ArcGIS Pro: 3.4.3). The KDE included GPS points from all deer with at least 10 days of GPS locations, for a total of 122 deer (99 female and 23 male). These included the 59 deer included in the disease transmission risk models, plus additional deer that were excluded from that analysis due to the reasons described above. All deer locations were pooled together, and we calculated a separate KDE surface for each year of data, defined as the one-year period between CWD testing events. The search radius for the KDE was set at 1,515 m, the average daily distance traveled by all deer. We then drew contours from 0–100% at 5% intervals as a measure of relative intensity of deer use, where values of 0 represent map pixels with the highest density of use points and values of 100 represent map pixels with the lowest density (S1 Fig.). We then incorporated KDE as a predictor variable in the risk models by extracting KDE values at each GPS point for each deer and calculating the average value for each deer and year, as a proxy for how much time deer spent in high versus low deer density areas. The risk model representing hypothesis 2 contained KDE as a predictor for probability of new CWD infection, plus any of the non-environmental variables tested in hypothesis 1 that we found to be significant.

### Hypothesis 3: habitat suitability hypothesis

Under the habitat suitability hypothesis (hypothesis 3), risk of CWD transmission was expected to be greater in habitats associated with high suitability and selection by mule deer. Hypotheses 2 (deer density) and 3 (habitat suitability) are similar in that they both connect risk of disease transmission to intensity of deer use of different habitats. In both cases, it is assumed that areas with greater frequency or intensity of use will result in increased likelihood and intensity of prion deposition by CWD-infected deer, as well as higher probability of CWD infection through social contact. However, the two hypotheses differ in that hypothesis 2 assumes that risk is more tightly connected to observed, recent deer presences on the landscape (as represented by the KDE), while hypothesis 3 assumes risk is correlated to environmental covariates related to habitat suitability and selection. To test the habitat suitability hypothesis (3), we developed resource selection functions (RSFs) to identify habitats associated with high suitability. The RSF analysis differs from the KDE analysis in that the high suitability areas identified by RSF are connected to specific habitat properties (e.g., proximity to agriculture) but may or may not be heavily used by deer, while the KDE "hotspot" areas are known to be areas heavily used by deer within a specific time frame but contain no information about relationships with habitat properties and could be more marginal habitats.

The RSF analysis used generalized linear models (GLMs) to predict probability of use based on environmental properties expected to attract or deter mule deer. The analysis compared used points (GPS collar locations) to a sample of available points (randomly selected points within the study area, which may or may not coincide with used points), with environmental properties extracted at used and available points as model covariates. Environmental properties expected to attract or deter use by mule deer included covariates related to five broad categories: presence of agriculture [22,50], presence of human development [50,51], presence of water sources [52], topographic properties [50,53], and vegetation cover [50,53] (refer to S2 File for a detailed list of covariates). Since a portion of the herd migrated to a separate range during summer and a portion remained on winter range year-round [2], separate models were constructed for three different migratory groups: nonmigratory deer, migratory deer on summer range, and migratory deer on winter range. Though migratory deer during winter and nonmigratory deer used the same geographic area, we constructed separate models since habitat selection criteria may be different for the two groups. Migratory status of each deer was determined by visually inspecting the GPS record for each individual. GPS points used in this analysis included data from collared deer with at least 3 months of recorded locations, the minimum duration required to reliably categorize a deer as migratory or nonmigratory, for a total of 110 deer (91 female and 19 male). These included the 59 deer included in the disease transmission risk models. Of the GPS-collared individuals investigated in this study, approximately 45–55% of animals migrated each year.

We investigated habitat use over a range of spatial scales by calculating the mean of habitat variables within different buffer sizes of each GPS point (using the exactextractr package in R [54]) for continuous value variables (e.g., heat load index), or calculating a Euclidean distance or distance decay function [55] at different scales for proximity variables (e.g., distance to primary road). The distance decay function describes how the influence of a feature dissipates as distance from the feature increases. In our case, testing distance decay at different scales for proximity variables such as distance to roads tested whether the effect of roads was influential only at close distances, or if influence was still present far from roads. The distance decay function took the form $\exp^{-d/a}$, where $d$ is the Euclidean distance and $a$ is the scale that determines the shape of the decay. We selected the optimal form and scale for each habitat predictor by constructing univariable GLMs using R software [41], and comparing models using $AIC_c$. Refer to S2 File for additional details on RSF methods. We used final models to predict relative probability of habitat use and applied the models spatially across the area of interest. The area of interest was determined by first defining the minimum convex polygon (MCP) around the deer GPS points and then extending out in all directions by the maximum distance traveled by any mule deer in the study, which was 48,310 m. We standardized predicted probabilities to obtain relative probabilities of use ranging from 0 to 1, and then reclassified these relative probabilities into 20 quantile bins. We incorporated RSF quantile as a predictor variable in the risk models by extracting RSF values at each GPS point for each deer and calculating the average value for each deer and year, as a proxy for how much time deer spent in high versus low suitability habitat. GPS points were matched to the correct RSF surface: e.g., GPS points from a nonmigratory deer in 2013 were extracted from the predicted RSF surface specific to nonmigratory deer in 2013.

We constructed a suite of GLMs to predict CWD transmission risk, containing multiple covariates related to habitat suitability including RSF quantile values as a direct measure of habitat suitability as well as individual habitat variables believed to either attract or deter use by mule deer (refer to Table 1). These individual habitat variables included covariates from some of the same categories that were included in the resource selection function (RSF) analysis: agriculture, human development, water sources, and vegetation cover.

### Hypothesis 4: prion persistence hypothesis

The prion persistence hypothesis (hypothesis 4) assumed risk of CWD transmission would be higher in areas where certain environmental properties could make infectious prions more likely to persist and remain available for uptake. We constructed a suite of GLMs containing covariates expected to affect prion persistence on substrates, including

**Table 1. Environmental variables used to model the risk of mule deer newly testing positive for chronic wasting disease (CWD) based on habitats visited by the deer. Variables fall within the categories: agriculture, human development, water sources, topography, soil, and vegetation. Variables included in risk models exploring hypothesis 3 were expected to affect risk by controlling habitat suitability (either attracting or deterring mule deer from using certain habitats), and variables included in risk models exploring hypothesis 4 were expected to affect risk by increasing or decreasing prion persistence in the environment. Spatial resolution of all data sets is 30m unless otherwise stated.**

| Environmental variable | Data source/citation | Used in models: habitat suitability (hypothesis 3) | Used in models: prion persistence (hypothesis 4) |
|---|---|---|---|
| **Agriculture** | | | |
| Distance to cropland in meters | USDA NASS CDL [56] | x | |
| Proportion of area designated as cropland | USDA NASS CDL [56] | x | |
| Distance to irrigated land in meters | IrrMapper [57] | x | |
| **Human development** | | | |
| Distance to primary road (i.e., interstate highways) in meters | TIGER [38] | x | |
| Distance to secondary road (i.e., state highways) in meters | TIGER [38] | x | |
| Density of secondary roads | TIGER [38] | x | |
| Distance to local or 4WD road in meters | TIGER [38] | x | |
| Percent time deer spent in developed (high, medium, and low intensity) | NLCD 2021 [49] | x | |
| **Water sources** | | | |
| Distance to perennial water source (combined spring/seep, lake/pond/reservoir, stream/river layers) in meters | USGS NHD [39] | x | x |
| Distance to ephemeral stream/river in meters | USGS NHD [39] | x | |
| Distance to intermittent lake/pond/reservoir in meters | USGS NHD [39] | x | |
| Percent time deer spent in wetland | NLCD 2021 [49] | x | x |
| **Topography** | | | |
| Elevation | From [58] | | x |
| Heat Load Index (HLI); a measure of incident solar radiation | From [58] | | x |
| Vector Ruggedness Measure (VRM); higher values = more rugged terrain | From [58] | | x |
| Compound Topographic Index (CTI); higher values = flatter, wetter | From [58] | | x |
| Aspect; standardized so value 0 = north-facing and 1 = south-facing | Derived from elevation [58] and standardized using methods from [59] | | x |
| **Soil** | | | |
| Surface clay content (%) | [60] (100m resolution) | | x |
| Surface sand content (%) | [60] (100m resolution) | | x |
| Surface pH | [60] (100m resolution) | | x |
| Surface organic carbon (SOC) | [60] (100m resolution) | | x |
| Soil moisture | [61] (300m resolution) | | x |
| **Vegetation** | | | |
| Distance to nearest trees (cover >1%) in meters | NLCD TCC 2016 [62] | x | |
| Percent tree cover | RCMAP [63] | x | |
| Percent shrub cover | RCMAP [63] | x | |
| Percent herbaceous cover | RCMAP [63] | x | |
| Annual vegetation biomass (pounds/acre) | RAP v3 [64] | x | |
| Perennial vegetation biomass (pounds/acre) | RAP v3 [64] | x | |
| Proportion of area designated as early growth conifer/pinyon-juniper (1–10% canopy cover) | LANDFIRE 2016 Existing Vegetation Type [65] and RCMAP [63] | x | |

covariates related to water sources, topography, and soil properties (refer to Table 1 for list of covariates). As in the models related to hypothesis 3, we extracted spatial covariate values at each GPS point and aggregated by deer and year to obtain a single value.

### Hypothesis 5: combination hypothesis

The combination hypothesis (hypothesis 5) assumed risk of CWD transmission would be affected by covariates related to all of the previously discussed mechanisms, including density of deer use points, habitat selection and suitability, and prion persistence on the landscape. The set of candidate spatial covariates included covariates from the top model from each of hypotheses 1, 2, 3, and 4.

### Model selection

For models representing hypotheses 1, 3, and 4, we first fit univariable models with each candidate covariate. Hypothesis 2 consisted of a single candidate covariate, KDE, and so this step was not necessary. We ranked univariable models by $AIC_c$ as a measure of the relative predictive ability of each covariate. To determine the combination of covariates to be included in multivariable models, we first tested for correlation between candidate covariates, and where correlation ≥ 0.6 we retained the most explanatory covariate with the lowest $AIC_c$ value. We then constructed a global model for each hypothesis that included all uncorrelated covariates, and used the "dredge" function from the "MuMIn" package [66] to select the best supported model containing combinations of covariates for each hypothesis. Finally, we selected the best overall model by comparing $AIC_c$ of the best model from each model set representing the five hypotheses: null hypothesis, deer density hypothesis, habitat suitability hypothesis, prion persistence hypothesis, and combination hypothesis.

## Results

Our dataset of mule deer for which there were sufficient GPS and CWD testing records to include in transmission risk models consisted of 59 deer. Of these, 22 were of genotype *F and 37 genotype SS, and 27 deer eventually tested positive for CWD during the study while 32 never tested positive (Table 2).

### Resource Selection Function (RSF)

Across all behavioral types, there was no evidence to exclude any variables from the model according to least absolute shrinkage and selection operator (LASSO), and therefore the global model was used to identify areas of predicted use by mule deer (refer to S2 File for details on RSF methods). Note that while global models were used for all three behavior categories, each global model contained different variables selected during the group-specific model selection process.

Table 2. Numbers of GPS-collared mule deer used in models predicting relative risk of a deer newly testing positive for chronic wasting disease based on habitat visited by the deer. Deer individuals were identified by genotype and were either homozygous for serine at codon 225 ("SS" genotype), or were heterozygous/homozygous for phenylalanine at codon 225 ("*F" genotype). All individuals included in models did not test positive for chronic wasting disease (CWD) on first capture. The total number of individuals in each genotype group is further broken down into individuals that never tested positive for chronic wasting disease for the duration of the study ("Remained CWD-") and those that had tested positive by the end of the study ("Became CWD+"). All individuals in this analysis are female.

| Genotype | Total individuals | Remained CWD- | Became CWD+ |
| --- | --- | --- | --- |
| *F | 22 | 21 | 1 |
| SS | 37 | 11 | 26 |
| Total | 59 | 32 | 27 |

The most influential characteristics associated with habitat selected by nonmigratory deer were higher shrub cover, lower tree cover, farther from primary roads, closer to irrigated land, closer to secondary roads, farther from ephemeral streams/rivers, and closer to cropland (S2 Table and S2 Fig.). Migratory deer on summer range selected for habitat with higher shrub cover, lower density of secondary roads, and farther from primary roads (S3 Table and S3 Fig.). Migratory deer on winter range selected habitat with higher shrub cover, lower line density of secondary roads, farther from primary roads, closer to irrigated land, closer to secondary roads, closer to trees, closer to cropland, and higher aspect (i.e., south-facing slopes; S4 Table and S4 Fig.).

We created maps displaying predictions from the best RSF model for each migratory group (nonmigratory deer, migratory summer range, and migratory winter range) and classified relative probabilities into ordinal 20 quantile bins (0–20, where 20 = most selected for). Since several of the predictors in the RSF model were year-specific, a separate map was created for each year. However, year-to-year variation in predictors was small, and therefore differences in RSF quantile maps between years were also small and unlikely to influence the results of risk models incorporating RSF quantile. Maps for 2014 can be found in S5 Fig.

**Hypothesis 1: null model**

In the set of null models (hypothesis 1) containing non-environmental covariates, we found that the top five models were within 2 $AIC_c$ units and therefore competitive [48]. We chose the most parsimonious of the top models to represent the null hypothesis, which was the model containing only genotype (S5 Table and S6 Table). All subsequent models also controlled for genotype.

**Hypothesis 2: deer density hypothesis**

In the model testing the deer density hypothesis (hypothesis 2), the 95% confidence interval for the estimate for the effect of KDE on transmission risk included zero (S7 Table), suggesting a lack of evidence that deer that spent more time in areas with high GPS point density had an increased or decreased risk of CWD transmission.

**Hypothesis 3: habitat suitability hypothesis**

Among multivariable models containing habitat suitability covariates (hypothesis 3), the best-supported model contained genotype, distance to perennial water source during summer, distance to cropland during winter, and distance to secondary road (year-round), and for all covariates 95% confidence intervals did not include zero (S8 Table and S9 Table, and S6 Fig.). The estimate for distance to perennial water source was negative, suggesting that risk was higher for deer that spent more time closer to perennial water during summer. Similarly, risk was predicted to be higher for deer that spent more time closer to secondary roads and farther away from cropland during winter.

**Hypothesis 4: prion persistence hypothesis**

Among models containing prion persistence covariates (hypothesis 4), the best-supported model contained genotype, distance to perennial water source during summer, compound topographic index (CTI) during summer, and CTI during winter (S10 Table and S11 Table, and S7 Fig.). Compound topographic index is a topographic wetness index calculated from slope and size of upslope area, where higher CTI values are interpreted as flatter and wetter than high values. The model estimates suggest that disease risk was higher for deer that spent more time in flatter/wetter areas during summer, but lower for deer that spent more time in flatter/wetter areas during winter (S7 Fig.). As in the best habitat suitability model, coefficient estimate for distance to perennial water during summer was negative, though the 95% confidence interval for this covariate included zero. The coefficient estimate for CTI during summer was positive, but negative for CTI during winter, and neither 95% confidence interval included zero.

### Hypothesis 5: combination hypothesis

The best model containing a combination of habitat suitability and prion persistence covariates (hypothesis 5) contained genotype, distance to perennial water source during summer, CTI during summer, CTI during winter, distance to cropland during winter, and distance to secondary road (year-round) and for all covariates 95% confidence intervals did not include zero (S12 Table and S13 Table). The direction of the relationships between risk and each covariate was consistent with the best-performing habitat suitability hypothesis model and prion persistence hypothesis model reported above.

### Best-performing CWD transmission risk model

When the top model from each of the five hypotheses were compared, the best-performing overall model was the combination hypothesis model, containing genotype and a combination of habitat suitability and prion persistence covariates described above (Table 3, Figs 2 and 3).

We then used the best overall model to predict relative disease transmission risk over the entire study area and create a risk map (Fig 4). Because covariate effects differed by season (summer/winter) and genotype (SS/*F), risk estimates also differ by season and genotype. One of the predictors, distance to cropland, varied by year, and so predictions of risk were year specific; Fig 4 shows predictions of risk specific to 2014, the final year of data collection. Risk maps for the best habitat suitability model and the best prion persistence model can be found in S8 Fig. and S9 Fig., respectively. Risk maps for 2014 are available as raster files within a publicly available data release [67].

## Discussion

We modeled the probability a mule deer contracted CWD within a one year period based on environmental properties of the habitat used by the deer during that time, testing five hypotheses about which disease transmission mechanisms may be dominant in this system: 1) null hypothesis, in which models tested correlations between risk and non-environmental factors only; 2) deer density hypothesis, which tested correlation between risk and density of deer use points on the landscape as measured by a kernel density estimate (KDE) analysis; 3) habitat suitability hypothesis, in which models tested correlations between risk and covariates related to habitat selection; 4) prion persistence hypothesis, in which models tested correlations between risk and environmental covariates related to the persistence of prions on the landscape; and 5) combination hypothesis, in which models included covariates related to all mechanisms described above. We found that the best overall model was the combination hypothesis model, and included environmental predictors related to

Table 3. Comparison of models predicting probability a mule deer newly tested positive for chronic wasting disease (CWD) in a one-year period. Models listed are the top models representing each of five hypotheses: 1) null hypothesis (non-environmental covariates), 2) deer density hypothesis (containing kernel density estimate as a covariate), 3) habitat suitability hypothesis (containing environmental covariates related to habitat suitability), 4) prion persistence hypothesis (containing environmental covariates related to prion persistence), and 5) combination hypothesis (containing environmental covariates related to both habitat suitability and prion persistence). K indicates the number of parameters in each model, and $AIC_c$ refers to Akaike's Information Criterion for small sample sizes.

| Model name | K | $AIC_c$ |
|---|---|---|
| Combination hypothesis model | 7 | 71.62 |
| Habitat suitability hypothesis model | 6 | 76.30 |
| Prion persistence hypothesis model | 6 | 77.53 |
| Null model (genotype only) | 2 | 88.27 |
| Deer density hypothesis model (kernel density estimate [KDE] only) | 3 | 88.64 |
| Resource selection function (RSF) quantile only | 3 | 89.17 |

**Fig 2. Standardized coefficient estimates with standard errors from top model predicting chronic wasting disease transmission risk for the South Converse Mule Deer Herd based on spatial properties of habitat used by deer.** Covariates included *PRNP* genotype (shows the effect of a deer being genotype SS compared to reference type *F), compound topographic index (CTI) during summer, CTI during winter, distance to cropland during winter, distance to perennial water source during summer, and distance to secondary road.

both habitat suitability and prion persistence, but not the deer density metric (KDE). This suggests that differences exist in CWD risk on the landscape, and mechanisms related to both prion persistence and patterns of habitat suitability are important in this system.

The model representing the null hypothesis in our analysis included *PRNP* genotype as the only predictor, and we found that probability of contracting CWD in a one-year period did not differ by migratory status, age, or observation year. We had expected differences between migratory and nonmigratory deer: migratory deer may have an increased chance of encountering prions on the landscape since they have larger home ranges and are exposed to more habitat types, or nonmigratory deer on smaller home ranges may have an increased chance of contracting CWD through increased probability of social contact or repeated exposure to habitats with high prion contamination. However, we found no evidence to support either expectation. We also found no evidence that probability of contracting CWD differed by age, suggesting that susceptibility to CWD infection was approximately constant across ages for deer of the ages observed in this study. Estimated ages ranged from 1.5–9.5 years; however, 84% of data points were ages 3.5–5.5, and it is possible that the lack of samples from very old and very young individuals prevented our models from detecting an effect of age on disease transmission risk. Numerous studies have noted differences in CWD prevalence in deer by age class [68–71]; our results do not refute these findings, since our study examined probability of new infection at different ages and not disease prevalence.

The model representing the deer density hypothesis included KDE as a predictor as well as genotype, but the confidence interval for KDE overlapped zero (S7 Table) and model support according to AICc was no better than the null model (Table 3), suggesting that deer which spent more time in areas more heavily used by deer (KDE "hotspots") were not more likely to develop CWD. In constructing this hypothesis, we assumed KDE hotspots represented areas of increased likelihood and intensity of prion deposition by CWD-infected deer and higher probability of CWD infection through social contact [72–75]. However, the KDE analysis may fail to accurately represent social contact and prion deposition rates, as we only had a small percentage of the herd marked with GPS collars, and the temporal duration of our study was only four years out of the several decades during which CWD-infected animals have been depositing prions on the landscape. Furthermore, we were not able to account for the effects of the two other CWD-susceptible species which are present in the study area and also interact with mule deer and deposit prions on the landscape: white-tailed deer (*Odocoileus*

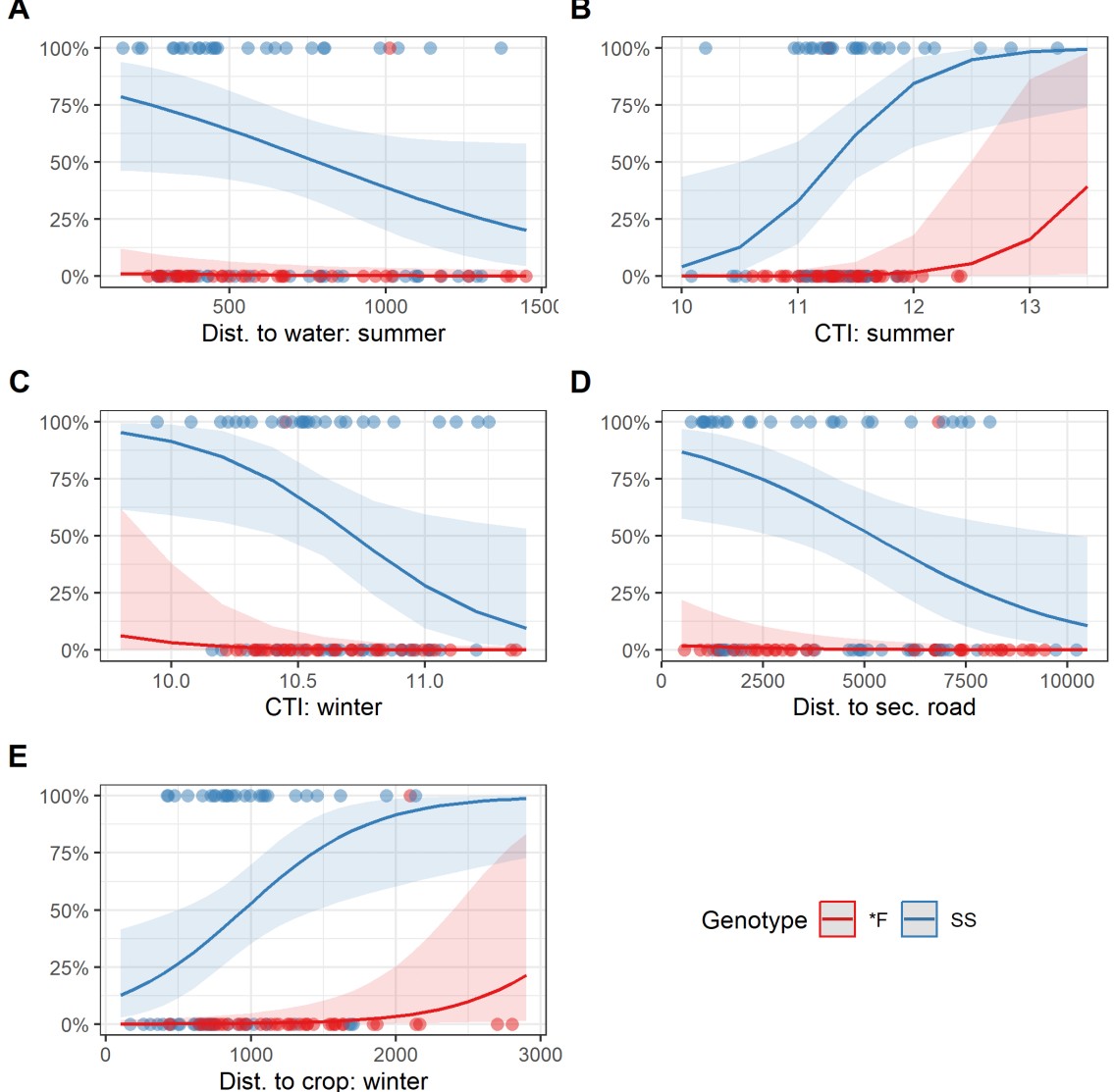

**Fig 3. Marginal effects of environmental covariates from top model predicting chronic wasting disease transmission risk for the South Converse Mule Deer Herd based on spatial properties of habitat used by deer.** Spatial covariates included A) distance to perennial water source during summer, B) compound topographic index (CTI) during summer, **C)** CTI during winter, D) distance to secondary road, and E) distance to cropland during winter. Color indicates differences in effects by genotype (*F or SS).

virginianus) and elk (*Cervus elaphus*) [1,76]. Though our model result fails to provide support for the hypothesis that CWD risk was related to mule deer density on the landscape, studies have shown that more detailed information about social contact (i.e., contact probabilities derived from proximity logging GPS collars) can be connected to CWD risk [5], and so we cannot conclusively determine whether host animal density was a relevant factor in this system.

The best risk model representing the habitat suitability hypothesis contained the habitat covariates distance to perennial water source during summer, distance to cropland during winter, and distance to secondary roads, but did not include the metric based directly on the resource selection function (RSF) quantiles. The finding that RSF quantile was not a sufficient predictor of risk suggests that deer spending more time in the most highly suitable habitat did not have a measurable

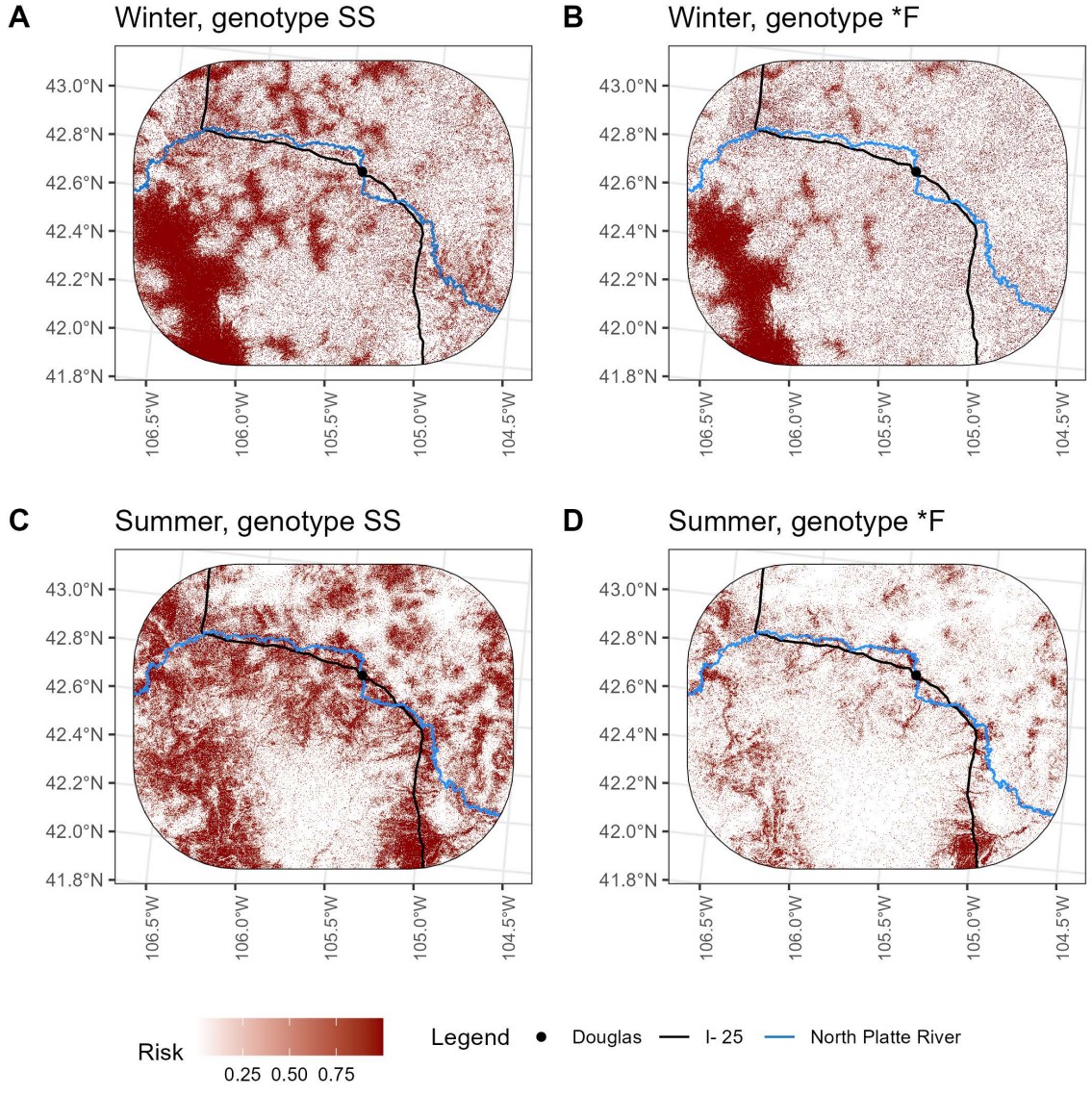

**Fig 4. Map showing relative risk of chronic wasting disease (CWD) transmission for mule deer in the South Converse Mule Deer Herd as predicted from top risk model.** Spatial predictors in top risk model included compound topographic index (CTI), distance to perennial water source, distance to cropland, and distance to secondary road. The black solid line bisecting the study area is Interstate 25. Since predictors are specific to season and deer genotype was also a predictor in the top model, maps are specific to genotype and season: A) genotype SS, winter, B) genotype *F, winter, C) genotype SS, summer, D) genotype *F, summer. Because one of the predictors, distance to cropland, varies by year, these maps are specific to data from 2014.

effect on probability of contracting CWD. However, as noted earlier, our RSF analysis identified areas that were highly suitable for the focal species, but these areas may or may not be used in practice. Resource selection function quantile was not strongly correlated to KDE at the pixel level (r = 0.1–0.4, across all migration status types and yearly KDEs), indicating that highly selected habitat was not strongly correlated to GPS point density. As we noted with the KDE metric, RSF quantiles may be an incomplete representation of the assumed mechanistic pathways of increased prion deposition and higher probability of social contact in highly suitable habitat, though the relationship may have been stronger if we had had data to identify the timing of CWD infection with more precision. However, direct comparisons of predictions from the RSF

analysis and the best habitat suitability risk model show parallels where specific elements of highly selected habitat coincided with increased risk, which supports the hypothesis that risk was higher in some highly selected habitats. All migratory groups (nonmigratory, migratory deer on summer range, and migratory deer on winter range) selected for areas near secondary roads. Our finding that CWD transmission risk was higher closer to secondary roads (Figs 2 and 3) supports the hypothesis that some areas highly selected by deer will be associated with higher risk, either through implied greater probability of social contact in these areas or increased rates of prion deposition by infected deer.

Discrepancies between predicted relationships of disease risk and habitat selection suggest that patterns of habitat selection alone do not fully explain risk. Shrub cover was among the strongest predictors in RSF models for nonmigratory and migratory deer in both seasons (S2-S4 Tables), which agrees with other studies of mule deer habitat selection that indicate preference for shrubland [50,53], but shrub cover was not identified as a top covariate in risk models. Conversely, distance to perennial water sources during summer was identified as an important covariate in risk models, but the effects of distance to perennial water sources in RSF models were small, and the direction of the relationship varied by migratory/nonmigratory status and by the form of the water source (lakes/reservoirs, streams/rivers, or springs/seeps). Additionally, all migratory groups selected for habitat close to cropland in the RSF analysis (refer to [50]), but risk models indicated decreased CWD risk close to cropland. Mismatches in magnitude and/or direction of environment-risk relationships and environment-selection relationships suggest that mechanisms related to prion deposition and persistence on the landscape, or other mechanisms independent of habitat selection, are apparent in this system [77–80].

The best risk model representing the prion persistence hypothesis contained the environmental variables distance to perennial water source during summer, and compound topographic index (CTI) during summer and winter. The risk model identified higher apparent risk in areas closer to perennial water sources during summer and at higher CTI values during summer. Distance to perennial water sources and CTI were also investigated as covariates in the RSF analysis, but had weak associations with habitat selection (Tables S2 and S4). Increased risk close to water sources during summer could be explained by the fact that animals in the clinical stage of CWD experience polydipsia/polyuria (excessive drinking/urination; [81,82]) and may therefore be attracted to water sources [83], where they deposit a substantial source of infectious prions from bodily fluids, feces, and ultimately their carcasses [84–87]. The finding that higher risk is associated with higher CTI during summer is consistent with the prion persistence hypothesis since prions can be transported by water [14] and would therefore collect in flat areas with large upslope catchments, the definition of high CTI values.

Risk models exploring the prion persistence hypothesis did not indicate connections between CWD risk and soil properties, including percent clay, despite literature evidence that soil clay content influences CWD transmission risk [24,25]. However, the clay content values found in our dataset represent a limited range, from 14–26% clay, compared to values ranging from 0–38% reported in [25] and 4.8–29.3% reported in [24], which both found a significant correlation between clay content and CWD risk. This illustrates one of the limitations of our study; though deer were tested annually, there remained high uncertainty in precise timing of CWD infection. Time between exposure to an infectious dose and the first positive test via tonsil biopsy can vary greatly: CWD can be detected as early as 42 days post-exposure [88], but in some cases infected deer may persist in a prolonged subclinical state for up to two years [89]. For the purpose of our models, we assumed that exposure most likely occurred within the 12-month period prior to the first positive test. Averaging over a year of habitat use points led to a greatly truncated range of environmental covariates representing apparent habitat use compared to the range of values available in the study area at the pixel level (S10 Fig.). This approach also averaged out nuances of individuals' habitat use during the one-year time frame. For example, one individual may spend the entire year on soils with moderate clay content while a second individual alternates between high and low clay content values, but both individuals would appear to use habitat the same average value according to our calculations. Furthermore, our model design would not have been able to identify small-scale sources of infection, such as a deer visiting a highly contaminated mineral lick [90]. Therefore, our models may not have been able to identify differences in probability

of contracting CWD based on fine-scale environmental properties, highlighting how the spatial scale of sources of CWD infection and risk influences interpreting the results of this study.

Another limitation of this study is that correlations among environmental covariates may obscure mechanistic relationships between environmental properties and disease risk. For example, apparent selection for secondary roads in the RSF analysis appears to contradict other studies which found that mule deer tended to avoid roads [50,51]. However, these studies did not differentiate between primary and secondary roads, a crucial difference in our study. Additionally, in our study area, distance to secondary road was correlated with other environmental properties that may be selected by deer. This effect has been observed in other systems; greater sage-grouse (*Centrocercus urophasianus*) were found to select habitats with a higher density of unpaved roads [91], but these roads are often associated with riparian areas which are likely the true habitat feature selected for by the animals. In our study area, distance to secondary road was correlated with distance to intermittent lake/pond/reservoir, elevation, soil moisture, and soil pH. Though deer selected for, and transmission risk was higher in, areas close to secondary roads, the mechanisms behind the two relationships may be connected to different, correlated environmental properties. Correlations could explain the findings that risk was higher farther from cropland and associated with lower CTI values during winter as well. Areas farther from cropland tended to be higher elevation and higher soil moisture, which may affect prion persistence on the landscape [14,15,24], and CTI was moderately correlated with heat load index and terrain roughness in our study area, either of which could influence habitat use by mule deer [50,92]. Further testing of these hypotheses can help identify mechanisms behind the observed relationships.

Though this study was aimed at detecting patterns of risk related to indirect transmission of CWD from environmental sources, the available data does not allow for complete disentanglement of direct and indirect disease transmission. Our models used differences in habitat use patterns by different individuals to infer differential risk, but it must be acknowledged that these individuals were also moving within the social structure of the herd, including both collared and uncollared individuals, and we were not able to explicitly account for these potential sources of animal-to-animal CWD transmission. Exposure to CWD through direct contact may be high during the rut [68], and it is hypothesized that the higher rates of CWD infection observed in males compared to females can be explained by differing social behavior during rut, when males fight with other males over territory and mate with multiple females [68]. Though we were prevented from including data from male mule deer in our risk models due to small sample size, our female-focused models were more likely to detect contributions of habitat use (and therefore indirect transmission) on the probability of a deer testing positive for CWD, since literature evidence suggests that males experience higher direct transmission risk than females [68]. Although our model results apply only to females, the patterns we have identified provide meaningful contributions to our understanding of CWD transmission on the landscape and can assist in guiding future research.

Previous modeling studies have made great advances in retrospectively understanding efficacy of different management strategies for controlling CWD [93,94] and simulating effects of management in emerging outbreaks [95]. Our effort extends previous work by explicitly considering infection from environmental sources, and specifically the fact that environmental risk varies based on habitat properties. Though our models were constructed to be predictive rather than explanatory [32], our results show observable differences in probability of a deer contracting CWD based on environmental characteristics, which allowed us to create maps for our area of interest of expected disease transmission risk from environmental factors. The risk map could be used to design CWD sampling and surveillance efforts, in conjunction with other local data. For example, combining our approach to identify potential environmental reservoirs of CWD with new methods currently being developed to detect CWD prions in soils [96] or use detection dogs to locate prion-infected feces in the field [97] may be able to detect the arrival of CWD in a new location earlier than by relying on opportunistic hunter-kill and found carcass testing alone, allowing managers to activate mitigation plans (e.g., culling, selective harvest) and increasing the chance of avoiding an outbreak.

While the relationships we identified are specific to a particular geographic area and may not be applicable in other regions with different habitat types, climates, and susceptible species present, our analysis can be used as a template for similar investigations. Future work can incorporate these maps into an individual based model (IBM) simulation framework, for example using HexSim software [98]. Simulations can test predictions about the rates of infection through direct and indirect routes of transmission, and simulate the impact of different management and harvest strategies on the progression of a CWD outbreak in a given population. This modeling tool can be used by state wildlife agencies to test efficacy of proposed harvest strategies aimed at reducing local CWD prevalence and spread.

## Supporting information

**S1 Fig. Map showing kernel density estimate (KDE) of density of deer GPS points from February 2013-February 2014.** The black solid line bisecting the study area is Interstate 25.
(PNG)

**S2 Fig. Marginal effects of predictor variables included in the resource selection model for nonmigratory mule deer.** The solid green line represents the relative probability of use, where higher values indicate stronger selection for a given habitat characteristic when all other variables are held at their mean. Shaded area represents 95% confidence interval of prediction.
(JPEG)

**S3 Fig. Marginal effects of predictor variables included in the resource selection model for migratory mule deer on summer range.** The solid green line represents the relative probability of use, where higher values indicate stronger selection for a given habitat characteristic when all other variables are held at their mean. Shaded area represents 95% confidence interval of prediction.
(JPEG)

**S4 Fig. Marginal effects of predictor variables included in the resource selection model for migratory mule deer on winter range.** The solid green line represents the relative probability of use, where higher values indicate stronger selection for a given habitat characteristic when all other variables are held at their mean. Shaded area represents 95% confidence interval of prediction.
(JPEG)

**S5 Fig. Predicted habitat suitability for mule deer in the South Converse Mule Deer Herd divided into 20 quantile bins.** A value of 20 indicates the most suitable habitat. Resource selection functions (RSFs) were fit separately for A) nonmigratory deer, B) migratory deer on summer range, and C) migratory deer on winter range. Predictions depend on year-specific spatial covariates; maps are shown based on data from 2014 (refer to S1 File for descriptions and citations for data used in modeling).
(PNG)

**S6 Fig. Marginal effects of covariates from best chronic wasting disease transmission risk model examining covariates related to habitat suitability.** Covariates include A) distance to perennial water source during summer, B) distance to cropland during winter, and C) distance to secondary road. Color indicates differences in effect by genotype (*F and SS).
(TIF)

**S7 Fig. Marginal effects of covariates from best chronic wasting disease transmission risk model examining covariates related to prion persistence.** Covariates include A) distance to perennial water source during summer, B)

compound topographic index (CTI) during summer, and C) CTI during winter. Color indicates differences in effect by genotype (*F and SS).
(TIF)

**S8 Fig. Maps of relative risk of chronic wasting disease (CWD) transmission for mule deer in the South Converse Mule Deer Herd, based on predictions from best model including effects from only habitat suitability covariates (distance to perennial water source during summer, distance to secondary road, distance to cropland during winter).** The black solid line bisecting the study area is Interstate 25. Maps are specific to genotype and season: A) genotype SS, winter, B) genotype *F, winter, C) genotype SS, summer, D) genotype *F, summer.
(PNG)

**S9 Fig. Maps of relative risk of chronic wasting disease (CWD) transmission for mule deer in the South Converse Mule Deer Herd, based on predictions from best model including effects of prion persistence (distance to perennial water source during summer, compound topographic index during summer and winter).** The black solid line bisecting the study area is Interstate 25. Maps are specific to genotype and season: A) genotype SS, winter, B) genotype *F, winter, C) genotype SS, summer, D) genotype *F, summer.
(PNG)

**S10 Fig. Ranges of environmental covariate values represented in the area of interest (pink lines), compared to the range of covariate values as implemented in risk models (blue lines; values extracted at deer GPS use points and averaged by deer and year).**
(TIF)

**S1 Table. Duration of GPS record (1–4 years) of each mule deer from which data was incorporated into disease transmission risk models.** Risk models used data from 59 female mule deer from the South Converse Mule Deer Herd. Each deer was equipped with a GPS collar which recorded locations every 6 or 8 hours for 1–4 years.
(PDF)

**S2 Table. The standardized parameters associated with each variable included in the global model depicting habitat use of nonmigratory mule deer (*Odocoileus hemionus*) of the South Converse Mule Deer Herd in south-central Wyoming.** Location data used in constructing the model was collected from 2010–2013 as part of a chronic wasting disease study. "Form" indicates the structure of the covariate. "Proportion" indicates the amount of a categorical variable within the specified moving window, refer to "scale;" "Mean" refers to the average value of a continuous covariate over the specified moving window, refer to "scale;" "Euclidian" is the straight-line distance to the nearest feature of the covariate of interest; and "Decay" is a distance metric calculated using a decay function where the effect diminishes as the distance from the feature increases. The rate of decay is specified under "scale." "β" and "SE" indicate each parameter estimate and standard error respectively. Note that positive coefficients associated with Euclidean distance measures indicate that predicted mule deer use is farther from the feature while negative coefficients indicate that predicted use is closer to the feature. The opposite relationship is true for decay distance measures.
(PDF)

**S3 Table. The standardized parameters associated with each variable included in the global model depicting habitat use of migratory mule deer (*Odocoileus hemionus*) while on their summer range of the South Converse Mule Deer Herd in south-central Wyoming.** Location data used in constructing the model was collected from 2010–2013 as part of a chronic wasting disease study. "β" and "SE" indicate each parameter estimate and standard error respectively. Note that positive coefficients associated with Euclidean distance measures indicate that predicted mule deer use is

farther from the feature while negative coefficients indicate that predicted use is closer to the feature. The opposite relationship is true for decay distance measures.
(PDF)

**S4 Table. The standardized parameters associated with each variable included in the global model depicting habitat use of migratory mule deer (*Odocoileus hemionus*) while on their winter range of the South Converse Mule Deer Herd in south-central Wyoming.** Location data used in constructing the model was collected from 2010–2013 as part of a chronic wasting disease study. "β" and "SE" indicate each parameter estimate and standard error respectively. Note that positive coefficients associated with Euclidean distance measures indicate that predicted mule deer use is farther from the feature while negative coefficients indicate that predicted use is closer to the feature. The opposite relationship is true for decay distance measures.
(DF)

**S5 Table. Comparison of generalized linear models predicting probability a mule deer newly tested positive for chronic wasting disease (CWD) in a one-year period based on non-environmental predictors only: *PRNP* genotype, deer age, observation year, and migratory/nonmigratory status.** K indicates the number of parameters in each model, and $AIC_c$ refers to Akaike's Information Criterion for small sample sizes.
(PDF)

**S6 Table. Parameter estimates with 95% confidence intervals from generalized linear model predicting probability a deer newly tested positive for chronic wasting disease in a one-year period, containing genotype as a covariate.**
(PDF)

**S7 Table. Parameter estimates from generalized linear model predicting probability a deer newly tested positive for chronic wasting disease in a one-year period, containing genotype and kernel density estimate (KDE) as covariates.**
(PDF)

**S8 Table. Covariate estimates for top 15 models according to Akaike's Information Criterion for small sample sizes ($AIC_c$) including combinations of multiple environmental variables related to habitat suitability.** *PRNP* genotype was included in all models. A covariate estimate of "NA" indicates the covariate was not included in the specified model. RCMAP refers to data from the Rangeland Condition Monitoring Assessment and Projection project (Rigge et al. 2024).
(PDF)

**S9 Table. Parameter estimates from top generalized linear model predicting probability a deer newly tested positive for chronic wasting disease in a one-year period, containing covariates related to habitat suitability.** Top model included genotype, distance to cropland during winter, distance to perennial water source during summer, and distance to secondary road. Continuous covariates were standardized prior to model fit.
(PDF)

**S10 Table. Covariate estimates for top 15 models according to Akaike's Information Criterion for small sample sizes ($AIC_c$) including combinations of multiple environmental variables related to prion persistence in the environment.** *PRNP* genotype was included in all models. A covariate estimate of "NA" indicates the covariate was not included in the specified model. CTI = Compound Topographic Index; HLI = Heat Load Index; SOC = Surface Organic Carbon; VRM = Vector Ruggedness Measure.
(PDF)

**S11 Table. Parameter estimates from top generalized linear model predicting probability a deer newly tested positive for chronic wasting disease in a one-year period, containing covariates related to prion persistence in the environment.** Top model included genotype, compound topographic index (CTI) during summer, CTI during winter, and distance to perennial water source during summer. Continuous covariates were standardized prior to model fit.
(PDF)

**S12 Table. Covariate estimates for top 15 models according to Akaike's Information Criterion for small sample sizes (AIC$_c$) including combinations of multiple environmental variables related to habitat suitability and prion persistence in the environment.** *PRNP* genotype was included in all models. A covariate estimate of "NA" indicates the covariate was not included in the specified model. CTI = Compound Topographic Index; KDE = Kernel Density Estimate; RSF = Resource Selection Function.
(PDF)

**S13 Table. Parameter estimates from top generalized linear model predicting probability a deer contracted chronic wasting disease in a one-year period, containing covariates related to both habitat suitability and prion persistence in the environment.** Top model included genotype, distance to cropland during winter, distance to secondary road, distance to perennial water source during summer, compound topographic index (CTI) during summer, and CTI during winter. Continuous covariates were standardized prior to model fit.
(PDF)

**S1 File. Spatial data details and processing steps.** This file includes additional details on sources of spatial data used as covariates in the resource selection function and risk model analyses. This file also describes the data processing steps in detail.
(PDF)

**S2 File. Resource selection function details.** This file includes additional details on methods and results of the resource selection function analysis done to support the manuscript "Modeling chronic wasting disease transmission risk in mule deer related to habitat characteristics.".
(PDF)

## Acknowledgments

We thank Matthew Holloran for contributing to project conception and grant writing during the planning stages of this project. Any use of trade, firm, or product names is for descriptive purposes only and does not imply endorsement by the U.S. Government.

## Author contributions

**Conceptualization:** Erica M Christensen, Nathan J Kleist, David R Edmunds, Julie A. Heinrichs, D Joanne Saher, Ashley L Whipple, Cameron L Aldridge.

**Data curation:** Erica M Christensen, Nathan J Kleist, D Joanne Saher, Ashley L Whipple, Melia DeVivo.

**Formal analysis:** Erica M Christensen, Nathan J Kleist, D Joanne Saher.

**Funding acquisition:** David R Edmunds, Julie A. Heinrichs, Cameron L Aldridge.

**Investigation:** David R Edmunds, Melia DeVivo.

**Methodology:** Erica M Christensen, Nathan J Kleist, D Joanne Saher.

**Supervision:** Julie A. Heinrichs.

**Visualization:** Erica M Christensen, Nathan J Kleist.

**Writing – original draft:** Erica M Christensen.

**Writing – review & editing:** Erica M Christensen, Nathan J Kleist, David R Edmunds, Julie A. Heinrichs, D Joanne Saher, Ashley L Whipple, Melia DeVivo, Cameron L Aldridge.

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
