## [Decision Letter · Decision Letter 0]

14 Sep 2025

Dear Dr. Christensen,

Thank you for submitting your manuscript to PLOS ONE. After careful consideration, we feel that it has merit but does not fully meet PLOS ONE’s publication criteria as it currently stands. Therefore, we invite you to submit a revised version of the manuscript that addresses the points raised during the review process.

We look forward to receiving your revised manuscript.

Kind regards,

Rodrigo Morales

Academic Editor

PLOS ONE

Journal Requirements:

3. In this instance it seems there may be acceptable restrictions in place that prevent the public sharing of your minimal data. However, in line with our goal of ensuring long-term data availability to all interested researchers, PLOS’ Data Policy states that authors cannot be the sole named individuals responsible for ensuring data access (http://journals.plos.org/plosone/s/data-availability#loc-acceptable-data-sharing-methods).

5. We note that Figure 1 in your submission contain map images which may be copyrighted. All PLOS content is published under the Creative Commons Attribution License (CC BY 4.0), which means that the manuscript, images, and Supporting Information files will be freely available online, and any third party is permitted to access, download, copy, distribute, and use these materials in any way, even commercially, with proper attribution. For these reasons, we cannot publish previously copyrighted maps or satellite images created using proprietary data, such as Google software (Google Maps, Street View, and Earth). For more information, see our copyright guidelines: http://journals.plos.org/plosone/s/licenses-and-copyright.

7. Please remove all personal information, ensure that the data shared are in accordance with participant consent, and re-upload a fully anonymized data set.

Additional guidance on preparing raw data for publication can be found in our Data Policy (https://journals.plos.org/plosone/s/data-availability#loc-human-research-participant-data-and-other-sensitive-data) and in the following article: http://www.bmj.com/content/340/bmj.c181.long....

Reviewers' comments:

Reviewer's Responses to Questions

**Comments to the Author**

1. Is the manuscript technically sound, and do the data support the conclusions?

Reviewer #1: Yes

Reviewer #2: Yes

2. Has the statistical analysis been performed appropriately and rigorously?

Reviewer #1: Yes

Reviewer #2: Yes

3. Have the authors made all data underlying the findings in their manuscript fully available?

Reviewer #1: Yes

Reviewer #2: No

4. Is the manuscript presented in an intelligible fashion and written in standard English?

Reviewer #1: Yes

Reviewer #2: Yes

Reviewer #1: Manuscript Number PONE-D-25-40590

Title: Modeling chronic wasting disease transmission risk related to habitat

Characteristics.

The authors have conducted a challenging study developing predictive models of CWD indirect transmission in mule deer using data from GPS-collared animals tested for CWD in Wyoming. Challenges associated with CWD and its impact on cervids continue to require an integration of field, laboratory, and biostatistical approaches to address disease expansion, persistence, surveillance, and control strategies.

This study modeled the probability of mule deer acquiring CWD (new infections) while accounting for variables that may impact indirect transmission of CWD. It accounted for the environmental properties of the study area that facilitate prion persistence, as well as for animal genotype, age, migratory status, and covariates associated with deer density and habitat characteristics used by the fifty-nine mule deer in the study. The best models did not include deer density, but they did include habitat suitability and environmental properties associated with prion retention. The study revealed differences in the risk of CWD in the landscape. Furthermore, PRNP genotype, but not age or migratory status, influenced the probability of contracting CWD, and habitat selection alone was unable to explain CWD risk in its entirety.

The manuscript was well written, provided detailed descriptions of the process, logical transitions, and a clear presentation of the mathematical models and results. I enjoyed reading the manuscript and going over the supplementary material.

My only recommendations are:

1) Add "mule deer" to the title.

2) Reference figures 2, 3, and 4 in the text.

3) To add to the figures (2,3 &4) or to the discussion/conclusion, an example of how wildlife managers can use this work. That is, help them understand how to apply this work. After all, the abstract concludes: with one of the applications of this work as these “spatially varying risk factors can support managers in designing data collection and disease management strategies.”

Reviewer #2: Review for PONE-D-25-40590

The authors utilize movement data from mule deer (tested annually) to understand Chronic Wasting Disease transmission risk from environmental sources. They approach this by addressing five competing hypotheses to explore different transmission pathways, with change in CWD-status as the dependent variable:

1) null hypothesis, disease risk is driven by demographic factors only (e.g., age, PRNP genotype),

2) deer density hypothesis (areas more highly used by deer, used KDE),

3) habitat suitability hypothesis, in which risk is related to covariates that affect habitat selection (used RSF),

4) prion persistence hypothesis, risk is driven by environmental covariates that affect the persistence of prions on the landscape,

5) combination hypothesis, in which covariates representing deer density, habitat suitability, and prion persistence are all important.

This study is relevant and warranted, given the spread of CWD and the implications of disease on many economies and properties. The question of environmental transmission has been an important one that governmental, academic, and private entities are interested in addressing, but often lack the data (e.g., change in infection status) to do so. This marks one of the first occasions where such a dataset is available to assess environmental characteristics.

Below we reply to PLOS One reviewer questions (as posted on their website) and we provide specific comments intended to increase clarity of the manuscript.

PLOS One review questions

1. What are the main claims of the paper and how significant are they for the discipline?

The authors identified that the best model in determining risk of CWD infection status was the combination hypothesis. They identified that significant factors in transmission risk included the individual’s genotype, distance to perennial water source during summer, CTI during summer, CTI during winter, distance to cropland during winter, distance to secondary road (year-round). Risk was higher for deer that spent more time near perennial water sources and secondary roads, and farther away from cropland during winter. Disease risk was also higher for deer that spent more time in flatter and wetter areas during winter, but lower for deer in flatter and wetter areas during summer.

2. Are the claims properly placed in the context of the previous literature? Have the authors treated the literature fairly?

Yes, they specifically explored environmental transmission risk, compared their findings with previous work (e.g., percent clay content) and expanded upon previous work by creating risk maps with several environmental factors contributing to infection risk. The authors should consider adding a limitations paragraph in their discussion (i.e., use of only female mule deer).

3. Do the data and analyses fully support the claims? If not, what other evidence is required?

Yes, the data and analyses support the claims.

4. PLOS One encourages authors to publish detailed protocols and algorithms as supporting information online. Do any particular methods used in the manuscript warrant such treatment? If a protocol is already provided, for example for a randomized controlled trial, are there any important deviations from it? If so, have the authors explained adequately why the deviations occurred?

Not applicable.

5. If the paper is considered unsuitable for publication in its present form, does the study itself show sufficient potential that the authors should be encouraged to resubmit a revised version?

It is suitable for publication with some edits.

6. Are original data deposited in appropriate repositories and accession/version numbers provided for genes, proteins, mutants, diseases, etc.?

Information not disclosed. Authors did not include a data sharing agreement.

7. Does the study conform to any relevant guidelines such as CONSORT, MIAME, QUORUM, STROBE, and the Fort Lauderdale agreement?

Not applicable.

8. Are details of the methodology sufficient to allow the experiments to be reproduced?

Not reproducible with the current details provided, however, suggestions to improve clarity and missing details are provided in the line-by-line comments below.

9. Is any software created by the authors freely available?

Not applicable.

10. Is the manuscript well organized and written clearly enough to be accessible to non-specialists?

The methods and results could use some additional formatting and rearrangement to help with clarity.

Major comments:

• Some of the methods section was difficult to follow for an unfamiliar reader. The authors might consider rearranging the methods to have the following sections: (1) Long-term monitoring, where the mule deer monitoring project is described, (2) separate sections to describe the approaches for each hypothesis, (3) model selection across hypotheses, and (4) methods for creation of the disease risk map.

o If authors pursue the aforementioned, reformatting the results in a similar way would help with flow.

• Methods. Citations needed for approaches, including moving window analysis, Euclidean distance, and Euclidean distance decay function; and for packages associated with the KDEs, RSFs, and moving window analysis.

• Results. An opening paragraph with overall results (e.g., how many mule deer were positive, how many were migratory, how many had the gene target you were looking at in your models) before the model results

Minor comments:

Ln 14-16: CWD can also be spread vertically

Ln 30. “Results of risk models also indicated increased risk in areas not associated with high resource selection, suggesting that infection risk was also elevated in areas where environmental properties facilitate prion retention” The “not associated with resource selection” does not provide context as to the features of these areas that may facilitate prion submission. If words permit, it would be useful to name a few.

Ln 110. There is some information in the introduction that could be moved to the Methods. Specifically, starting at “Hypotheses 2 […]” through Ln 125 (the entire end of this paragraph)– this content feels like it can be moved to the methods. I understand the desire to disclose this information early on, as I also wondered how H2 and H3 differed, but perhaps a brief statement that model differences and descriptions are outlined in the methods is sufficient.

Ln 125. A transition into the methods would help with flow.

Ln 132. What portion of the population migrated to the summer range?

Ln 138-139. Can you please provide a reference for deer aging based on tooth wear. Also, what was the decontamination protocol for the jaw spreader (or equivalent)?

Ln 147. Only females included in the study – how is this placed in the context of other literature (M v F) exploring environmental influences of spread and risk of infection?

Ln 149-152. Rigorous filtering was implemented for inclusion in the data set. Each animal needed one-year of GPS that had points pre-post CWD positive, collar had to have consistent reads (no gap in data), and at least 6 lymphoid follicles tested.

o What collar brands were used?

o How many GPS points were required pre- and post- CWD test?

Ln 156. Reference to Table 01 should be in the methods

Ln 164- 165. The authors state “expected to attract or deter use by mule deer”, could you please provide citations?

Ln 168. This is the first use of “RSF” but it has not yet been spelled out with the acronym.

Ln 201. Classifying migratory status as behavior is a bit confusing, can you use different terminology here? You use “migratory groups” in the Discussion (Ln 439-440), this might be a better description.

Ln 217-219. GLMs are defined earlier so no need to write out here. Should also bump the R package citation to earlier.

Ln 286. It might be helpful to reiterate that behavior types corresponds to migratory groups.

Discussion. Were any discrepancies observed between your results and previously published literature (e.g., soil clay content) possibly influenced by the limitations associated with the individual animals – specifically, do you think these results may be limited since only females are used? Could this explain some differences observed here relative to past work (are they directly comparable)? Please add a few sentences discussing the limitations of using only females.

Table 1. Additional information is needed in this table caption for it to be stand-alone table (also may fit better in the result section)

Table 2. Are the column labels supposed to reflect the hypotheses? The language used in-text earlier in the manuscript made it seem like RSF models were used to test habitat suitability hypothesis (Ln 119-120), so what are the columns representing (which model?) It is assumed that the “risk model: prion” is to address hypothesis 4 (prion persistence)?

We are grateful for the opportunity to review this manuscript and applaud the authors on their excellent work.

Alynn Martin & Ashlyn Halseth-Ellis

.

Reviewer #1: No

Reviewer #2: **Yes:** Alynn MartinAlynn MartinAlynn MartinAlynn Martin

While revising your submission, please upload your figure files to the Preflight Analysis and Conversion Engine (PACE) digital diagnostic tool, https://pacev2.apexcovantage.com/. PACE helps ensure that figures meet PLOS requirements. To use PACE, you must first register as a user. Registration is free. Then, login and navigate to the UPLOAD tab, where you will find detailed instructions on how to use the tool. If you encounter any issues or have any questions when using PACE, please email PLOS at . PACE helps ensure that figures meet PLOS requirements. To use PACE, you must first register as a user. Registration is free. Then, login and navigate to the UPLOAD tab, where you will find detailed instructions on how to use the tool. If you encounter any issues or have any questions when using PACE, please email PLOS at . PACE helps ensure that figures meet PLOS requirements. To use PACE, you must first register as a user. Registration is free. Then, login and navigate to the UPLOAD tab, where you will find detailed instructions on how to use the tool. If you encounter any issues or have any questions when using PACE, please email PLOS at . PACE helps ensure that figures meet PLOS requirements. To use PACE, you must first register as a user. Registration is free. Then, login and navigate to the UPLOAD tab, where you will find detailed instructions on how to use the tool. If you encounter any issues or have any questions when using PACE, please email PLOS at figures@plos.org. Please note that Supporting Information files do not need this step.. Please note that Supporting Information files do not need this step.

---

## [Author Response · Author response to Decision Letter 1]

5 Feb 2026

PONE-D-25-40590

Modeling chronic wasting disease transmission risk in mule deer related to habitat characteristics

PLOS ONE

Journal Requirements:

We have adjusted the manuscript and S1 Supplement to meet style requirements.

No original data were collected for this study, so there is no study-specific IRB or ethics committee information to report. Data collection methods and approvals were reported in the original study publication (DeVivo et al. 2017), which we have repeated in the Methods section of our manuscript for informational purposes.

- DeVivo MT, Edmunds DR, Kauffman MJ, Schumaker BA, Binfet J, Kreeger TJ, et al. Endemic chronic wasting disease causes mule deer population decline in Wyoming. PLOS ONE. 2017;12: e0186512. doi:10.1371/journal.pone.0186512

3. In this instance it seems there may be acceptable restrictions in place that prevent the public sharing of your minimal data. However, in line with our goal of ensuring long-term data availability to all interested researchers, PLOS’ Data Policy states that authors cannot be the sole named individuals responsible for ensuring data access (http://journals.plos.org/plosone/s/data-availability#loc-acceptable-data-sharing-methods).

We have added a citation to a portion of the data which is stored on Dryad: “Data on each deer’s sex, genotype, age at first capture, CWD test result at first capture, date of last test with CWD not detected, and date of first positive CWD test result (if applicable) are publicly available on Dryad (DeVivo et al. 2018).” (line 170 in tracked changes document).

The GPS locations are sensitive information and cannot be made publicly available, but the data owner has uploaded the data to the Movebank repository. We have added the following sentence to the methods citing this: “The GPS locations are available by request from Movebank (movebank.org, study name “Chronic Wasting Disease Ecology and Epidemiology of Mule Deer in Wyoming,” study ID 7727287671).” (line 173 in tracked changes document).

DeVivo MT, Edmunds DR, Kauffman MJ, Schumaker BA, Binfet J, Kreeger TJ, et al. Data from: Endemic chronic wasting disease causes mule deer population decline in Wyoming. Dryad; 2018. p. 51623 bytes. doi:10.5061/DRYAD.H66CN

We have created a USGS data release where the final raster maps resulting from these analyses are publicly available. We have added the citation to this data release (line 653 in tracked changes document, the data can be found here: doi.org/10.5066/P1K3QFC8).

5. We note that Figure 1 in your submission contain map images which may be copyrighted. All PLOS content is published under the Creative Commons Attribution License (CC BY 4.0), which means that the manuscript, images, and Supporting Information files will be freely available online, and any third party is permitted to access, download, copy, distribute, and use these materials in any way, even commercially, with proper attribution. For these reasons, we cannot publish previously copyrighted maps or satellite images created using proprietary data, such as Google software (Google Maps, Street View, and Earth). For more information, see our copyright guidelines: http://journals.plos.org/plosone/s/licenses-and-copyright. We require you to either (a) present written permission from the copyright holder to publish these figures specifically under the CC BY 4.0 license, or (b) remove the figures from your submission:

a. You may seek permission from the original copyright holder of Figure 1 to publish the content specifically under the CC BY 4.0 license. We recommend that you contact the original copyright holder with the Content Permission Form (http://journals.plos.org/plosone/s/file?id=7c09/content-permission-form.pdf) and the following text: “I request permission for the open-access journal PLOS ONE to publish XXX under the Creative Commons Attribution License (CCAL) CC BY 4.0 (http://creativecommons.org/licenses/by/4.0/). Please be aware that this license allows unrestricted use and distribution, even commercially, by third parties. Please reply and provide explicit written permission to publish XXX under a CC BY license and complete the attached form.” Please upload the completed Content Permission Form or other proof of granted permissions as an ""Other"" file with your submission. In the figure caption of the copyrighted figure, please include the following text: “Reprinted from [ref] under a CC BY license, with permission from [name of publisher], original copyright [original copyright year].”

b. If you are unable to obtain permission from the original copyright holder to publish these figures under the CC BY 4.0 license or if the copyright holder’s requirements are incompatible with the CC BY 4.0 license, please either i) remove the figure or ii) supply a replacement figure that complies with the CC BY 4.0 license. Please check copyright information on all replacement figures and update the figure caption with source information. If applicable, please specify in the figure caption text when a figure is similar but not identical to the original image and is therefore for illustrative purposes only. The following resources for replacing copyrighted map figures may be helpful: USGS National Map Viewer (public domain): http://viewer.nationalmap.gov/viewer/; The Gateway to Astronaut Photography of Earth (public domain): http://eol.jsc.nasa.gov/sseop/clickmap/; Maps at the CIA (public domain): https://www.cia.gov/library/publications/the-world-factbook/index.html and https://www.cia.gov/library/publications/cia-maps-publications/index.html NASA Earth Observatory (public domain): http://earthobservatory.nasa.gov/ Landsat: http://landsat.visibleearth.nasa.gov/ USGS EROS (Earth Resources Observatory and Science (EROS) Center) (public domain): http://eros.usgs.gov/# Natural Earth (public domain): http://www.naturalearthdata.com/

Figure 1 does not contain any copyrighted images. It was generated entirely by the authors using R, using publicly available data for elevation, U.S. state outlines, and locations of cities, rivers, and roads. We have added citations to data sources and software packages to the Figure 1 caption: “Data for elevation, road, and river were obtained from publicly available data sources (U.S. Census Bureau 2019, U.S. Geological Survey 2022, U.S. Geological Survey 2018). Map was created using R software (R Core Team 2025) including packages ‘ggplot2,’ ‘ggspatial,’ ‘maps,’ ‘cowplot,’ and ‘terra’ (Becker et al. 2025, Wickham 2016, Hijmans 2025, Dunnington 2025, and Wilke 2025).”

U.S. Census Bureau. TIGER/Line Shapefiles. 2019. Available: https://www.census.gov/geographies/mapping-files/time-series/geo/tiger- geodatabase-file.2019.html

U.S. Geological Survey. USGS National Hydrography in FileGDB 10.1 format. 2022. Available: https://www.sciencebase.gov/catalog/item/5ea068ae82cefae35a12a120

U.S. Geological Survey. 1/3rd arc-second digital elevation models (DEMs) - USGS National Map 3DEP downloadable data collection. 2018. Available: https://www.usgs.gov/core-science-systems/ngp/3dep/about-3dep-products- services

R Core Team. R: A language and environment for statistical computing. Vienna, Austria: R Foundation for Statistical Computing; 2024. Available: https://www.R-project.org/

Becker RA, Wilks AR, Brownrigg R, Minka TP, Deckmyn A. maps: Draw Geographical Maps. 2025. doi:10.32614/CRAN.package.maps

Wickham H. ggplot2: Elegant Graphics for Data Analysis. Springer-Verlag New York; 2016. Available: https://ggplot2.tidyverse.org

Hijmans RJ. terra: Spatial Data Analysis. 2025. doi:10.32614/CRAN.package.terra

Dunnington D. ggspatial: Spatial Data Framework for ggplot2. 2025. doi:10.32614/CRAN.package.ggspatial

Wilke CO. cowplot: Streamlined Plot Theme and Plot Annotations for ggplot2. 2025. doi:10.32614/CRAN.package.cowplot

We have renamed the Supplement to “S1 Supplement” and renamed supplemental tables and figures (e.g., “Table A1 in S1 Supplement”) in order to conform to the guidelines. We have also added the captions for figures and tables from the S1 Supplement at the end of the manuscript.

7. Please remove all personal information, ensure that the data shared are in accordance with participant consent, and re-upload a fully anonymized data set. Note: spreadsheet columns with personal information must be removed and not hidden as all hidden columns will appear in the published file. Additional guidance on preparing raw data for publication can be found in our Data Policy (https://journals.plos.org/plosone/s/data-availability#loc-human-research-participant-data-and-other-sensitive-data) and in the following article: http://www.bmj.com/content/340/bmj.c181.long.

We have ensured that no personal information appears in the submitted files.

The reviewers have not indicated any specific citation recommendations.

We have reviewed the reference list for completeness and to conform to formatting guidelines. We have not identified any retracted articles.

Reviewers' comments:

Reviewer's Responses to Questions

Comments to the Author

1. Is the manuscript technically sound, and do the data support the conclusions?

Reviewer #1: Yes

Reviewer #2: Yes

2. Has the statistical analysis been performed appropriately and rigorously?

Reviewer #1: Yes

Reviewer #2: Yes

3. Have the authors made all data underlying the findings in their manuscript fully available?

Reviewer #1: Yes

Reviewer #2: No

4. Is the manuscript presented in an intelligible fashion and written in standard English?

Reviewer #1: Yes

Reviewer #2: Yes

5. Review Comments to the Author

Reviewer #1: Manuscript Number PONE-D-25-40590

Title: Modeling chronic wasting disease transmission risk related to habitat

Characteristics.

The authors have conducted a challenging study developing predictive models of CWD indirect transmission in mule deer using data from GPS-collared animals tested for CWD in Wyoming. Challenges associated with CWD and its impact on cervids continue to require an integration of field, laboratory, and biostatistical approaches to address disease expansion, persistence, surveillance, and control strategies.

This study modeled the probability of mule deer acquiring CWD (new infections) while accounting for variables that may impact indirect transmission of CWD. It accounted for the environmental properties of the study area that facilitate prion persistence, as well as for animal genotype, age, migratory status, and cova

---

## [Decision Letter · Decision Letter 1]

16 Mar 2026

Modeling chronic wasting disease transmission risk in mule deer related to habitat characteristics

PONE-D-25-40590R1

Dear Dr. Christensen,

We’re pleased to inform you that your manuscript has been judged scientifically suitable for publication and will be formally accepted for publication once it meets all outstanding technical requirements.

Kind regards,

Rodrigo Morales

Academic Editor

PLOS One

Additional Editor Comments (optional):

While one author accepted the current version of the manuscript, the second pointed just minor comments that should be addressed before publication.

Reviewers' comments:

Reviewer's Responses to Questions

**Comments to the Author**

Reviewer #1: All comments have been addressed

Reviewer #2: All comments have been addressed

2. Is the manuscript technically sound, and do the data support the conclusions?

Reviewer #1: Yes

Reviewer #2: Yes

3. Has the statistical analysis been performed appropriately and rigorously?

Reviewer #1: Yes

Reviewer #2: Yes

4. Have the authors made all data underlying the findings in their manuscript fully available?

Reviewer #1: Yes

Reviewer #2: No

5. Is the manuscript presented in an intelligible fashion and written in standard English?

Reviewer #1: Yes

Reviewer #2: Yes

Reviewer #1: The authors did an excellent job at addressing the comments and suggestions of the reviewers. Thank you. We appreciate your efforts.

I have minor suggestions.

Recommendations:

LN 144: Odd text “progression than the”. Check the sentence. “… test positive for CWD and for those individuals that do acquire CWD, they experience delayed disease progression than the “SS” type”.

Ln 170: For the following sentence please cite the original study “We included only female deer in these models, as females were the main target of the original study and sample size of males was too small for analyses.”

Ln 194: Add the reference for the National Land Cover Database [NLCD] dataset used –

Ln 114, 139, 220: Genes are italicized. Therefore, here and throughout the manuscript (including supplemental files) write “PRNP” in Italics.

Ln 515-521: in the discussion it would be good to see a small note about the impact for the habitat suitability findings, of the statement in LN 379 “…several predictors in the RSF model were year-specific…” could the yearly variations have impacted the ability to see measurable effects?

Reviewer #2: The authors have addressed all of my comments, and I have no further feedback. I thank the authors for the opportunity to review their work, and believe it will be a significant contribution to the literature.

-Alynn Martin

.

Reviewer #1: No

Reviewer #2: No

---

## [Editor Report · Acceptance letter]

PONE-D-25-40590R1

PLOS One

Dear Dr. Christensen,

I'm pleased to inform you that your manuscript has been deemed suitable for publication in PLOS One. Congratulations! Your manuscript is now being handed over to our production team.

Kind regards,

on behalf of

Dr. Rodrigo Morales

Academic Editor

PLOS One